# Effects of Endurance and Endurance–Strength Training on Endothelial Function in Women with Obesity: A Randomized Trial

**DOI:** 10.3390/ijerph16214291

**Published:** 2019-11-05

**Authors:** Marzena Ratajczak, Damian Skrypnik, Paweł Bogdański, Edyta Mądry, Jarosław Walkowiak, Monika Szulińska, Janusz Maciaszek, Matylda Kręgielska-Narożna, Joanna Karolkiewicz

**Affiliations:** 1Department of Biology & Anatomy, Poznan University of Physical Education, 61-871 Poznan, Poland; 2Department of Treatment of Obesity, Metabolic Disorders & Clinical Dietetics, Poznan University of Medical Sciences, 60-569 Poznan, Poland; damian.skrypnik@gmail.com (D.S.); pbogdanski@ump.edu.pl (P.B.); mszulinska1@wp.pl (M.S.); matylda-kregielska@wp.pl (M.K.-N.); 3Department of Physiology, Poznan University of Medical Sciences, 60-781 Poznan, Poland; edytamadry@poczta.onet.pl; 4Department of Pediatric Gastroenterology & Metabolic Diseases, Poznan University of Medical Sciences, 60-572 Poznan, Poland; jarwalk@ump.edu.pl; 5Department of Physical Activity Science & Health Promotion, Poznan University of Physical Education, 61-871 Poznan, Poland; jmaciaszek@awf.poznan.pl; 6Department of Food & Nutrition, Poznan University of Physical Education, 61-871 Poznan, Poland; karolkiewicz@awf.poznan.pl

**Keywords:** physical training, lipid metabolism, skeletal muscle mass, atherosclerosis

## Abstract

Some investigations have demonstrated that a combined endurance–strength training is the most effective in the treatment of obesity. The aim of the research was to access how different trainings influence: endothelial function, lipid metabolism, and risk of atherosclerosis in women with obesity. In a randomized trial, 39 obese women aged 28–62 completed endurance (n = 22, 60-80% HR_max_) or combined training (n = 17, 20 minutes of strength exercises, 50–60% 1RM and 25 minutes of endurance training, 60-80% HR_max_). Before and after the intervention vascular endothelial function (endothelial nitric oxide synthase (eNOS), vascular endothelial growth factor (VEGF), thiobarbituric acid reactive substances (TBARS), blood total antioxidant capacity (TAC)), total cholesterol, low-density lipoprotein cholesterol (LDL-C), high-density lipoprotein cholesterol (HDL-C), triglycerides and C-reactive protein (CRP)as well as visceral adiposity index (VAI), total-body skeletal muscle mass and atherogenic index of plasma (AIP) were determined. After the trainings, in both groups total cholesterol and total-body skeletal muscle mass increased (*p* < 0.05). In the group undergoing combined training, lower (*p* < 0.05) VAI, AIP, CRP and LDL-C were noted. In the group undergoing endurance training TBARS concentration decreased (*p* < 0.01), while the HDL-C (*p* < 0.01) concentration as well as eNOS (*p* < 0.05) activity increased. No significant differences between groups were found, either before or after the programs. Both training programs led to the improvement of lipid metabolism, but only endurance training alone favorably changed indicators of endothelial functions in women with obesity.

## 1. Introduction

Obesity is associated with several established cardiovascular risk factors, including insulin resistance, dyslipidemia, hypertension, and chronic low-grade inflammation; together, these factors are damaging to the endothelium [1]. In addition to the known risk factors, there is now a growing body of evidence to support the role of several non-traditional risk factors as potential modulators of the endothelial phenotype in obesity, including pro-atherogenic adipokines, oxidative stress, and chronic inflammation [2].

Endothelial dysfunction represents the earliest abnormality in the development of vascular disease, and is pathophysiologically linked to the progression of atherosclerosis. Loss of endothelium-dependent vasodilator activity has been associated with reduced synthesis and availability of nitric oxide (NO)—a potent vascular smooth muscle relaxant [3]. The availability of NO in the plasma can be used as a measure of endothelial dysfunction but, due to its very short half-life, the measure most commonly used is activity of endothelial NO synthase [4].

Regular physical exercise can be a good stimulus for healthy cardiac function, probably due to shear stress-mediated mechanotransduction, which transforms mechanical forces into molecular changes and, in turn, modifies endothelial function and blood pressure. Chronic mechanical forces directly affect endothelial cell morphology, metabolism and inflammatory phenotype through signal transduction, and gene and protein expression [5].

Short-term training (<12 weeks) undertaken at moderate intensity (50% VO2max) for 30 minutes, five to seven times per week, has been demonstrated to increase endothelium-dependent vasodilatation in multiple patient populations (reviewed by Green et al. [6]). Exercise involving large muscle groups cause endothelium-dependent improvements in vasodilatation throughout the entire body. Adaptive changes in endothelial function were shown to be associated with physical training, depending (to a large extent) on the activity of endothelial nitric oxide synthase (eNOS) [4] and changes in oxidant/antioxidant balance [7].

Traditional endurance training is a recognized tool in the treatment of obesity. Several meta-analyses investigated the effects of aerobic exercise training on anthropometric and cardiometabolic risk factors. Reductions in body mass index (BMI), body weight, waist circumference, visceral adipose tissue, and increasing HDL-cholesterol (HDL-C) were observed after at least 12 weeks of exercise, applied three times per week, at intensity 53% to 85% of VO_2_max [8,9,10]. In recent years attention has also been paid to the beneficial effect of strength training, applied usually three days a week for at least 16 weeks. With respect to strength training, meta-analyses reported reductions of glycated haemeglobin, C-reactive protein (CRP), systolic blood pressure (SBP) and diastolic blood pressure (DBP) [11]. In the treatment/prevention of overweight/obesity and type 2 diabetes mellitus, combined training, containing both strength and endurance components has been proved to be most effective [12].

The nature of training may be important in inducing different shear stress patterns in blood vessels that affect endothelial function [13]. To our knowledge, no studies have assessed the effect of endurance–strength training on endothelial function and atherosclerosis risk in women with obesity. As such, the aim of the research presented herein was to compare the efficacy of three months of endurance training alone and combined endurance–strength training on endothelial function, lipid metabolism, and risk of atherosclerosis in women with obesity. It is possible that the application of varied physical effort by diversifying the stimuli in the form of different shear stress patterns can have a different impact on the studied indicators compared with endurance training alone.

## 2. Materials and Methods

### 2.1. Study Participants

Informed consent was obtained from all participants, and the study was approved by the Ethics Committee of Poznan University of Medical Sciences (case no. 1077/12; supplement no. 753/13). The study conformed to all ethical issues included in the Helsinki Declaration. 

Of the 163 women with obesity who were screened at the outpatient clinic of the Department of Internal Medicine, Metabolic Disorders and Hypertension (Poznan University of Medical Sciences, Poland), 44 women were enrolled. Five woman from combined training group were lost to follow up. Two women resigned from the study during the intervention due to personal reasons, three more were excluded from analysis due to low frequency or too young age in comparison to other women.

The inclusion criteria were as follows: age: 18 to 65 years; simple obesity (i.e., body mass index (BMI) ≥ 30 kg/m^2^); waist circumference > 80 cm; body fat ≥ 33%; and stable body weight in the month prior to the trial (permissible deviation ± 1 kg).

The exclusion criteria were as follows: secondary form of obesity and/or secondary form of hypertension; diabetes mellitus; history of coronary artery disease; stroke; congestive heart failure; clinically significant arrhythmias or conduction disorders; malignancy; use of dietary supplements within the 3 months prior to the study; poorly controlled hypertension (SBP > 140 mmHg and/or DBP > 90 mmHg), and/or modifications to antihypertensive treatment; lipid disorders requiring the implementation of drug treatment; clinically significant abnormalities in liver, kidney or thyroid gland function; clinically significant acute or chronic inflammatory process within the respiratory, digestive or genitourinary tracts, or the oral cavity, pharynx or paranasal sinuses; or, presence of connective tissue disease or arthritis; history of infection within the month prior to the study; nicotine, alcohol or drug abuse; and/or any other condition that, in the opinion of the investigators, would make participation either not in the best interest of the participant or could prevent, limit, or confound the efficacy of the study. 

### 2.2. Study Design

The study was designed as a prospective randomized trial. Participants were divided randomly (using a randomization list), based on the type of training, into one of two groups: endurance training group (Group E) or combined training group (Group C). Both groups underwent a 3-month physical training program. Outside the implemented program, all participants were instructed to maintain their normal physical activity, diet and not to use any dietary supplements. Dietary intake was assessed using interviews conducted at baseline and after completion of the trial. The amount of nutrients in participant’s daily diet was processed and evaluated using a dietetics computer program Dieta 5.0 (IŻŻ, 2011). The intake of nutrients, total caloric intake during the study were constant and comparable between the groups. Anthropometric parameters and physical capacity were measured and blood samples were taken at baseline and after completion of the physical training program. Detailed measurements of physical capacity and anthropometric parameters including body composition are described in our previous paper [14].

### 2.3. Anthropometric Parameters

Anthropometric measurements were conducted with the subjects wearing light clothing and no shoes. Weight was measured to the nearest 0.1 kg and height to the nearest 0.5 cm. BMI was calculated as weight divided by height squared (kg/m^2^). Obesity was defined as BMI ≥ 30 kg/m^2^. Waist circumference (cm) was measured at the level of the iliac crest at the end of normal expiration.

### 2.4. Visceral Adiposity Index and Total-Body Skeletal Muscle Mass

Visceral adiposity index (VAI) was calculated using the formula developed by Amato et al. [15]:(1)$$\mathrm{Females}:\mathrm{VAI}=\left(\frac{\mathrm{waist}\;\mathrm{circumference}}{36.58+\left(1.89\;\times \;\mathrm{BMI}\right)}\right)\times \left(\frac{\mathrm{TG}}{0.81}\right)\times \left(\frac{1.52}{\mathrm{HDL}}\right)$$

Total-body skeletal muscle mass was calculated using the skeletal muscle-prediction model created by Kim et al. [16]. The total-body skeletal muscle mass model is based on appendicular lean soft tissue (ALST), evaluated using dual-energy X-ray absorptiometry (DXA) and sex (0 = female; 1 = male). 

Total-body skeletal muscle mass = (1.13 × ALST) − (0.02 × age) + (0.61 × sex) + 0.97

Body composition was analyzed using dual-energy X-ray absorptiometry (DXA; GE Healthcare Lunar Prodigy Advance, GE Medical Systems, Milan, Italy). 

### 2.5. Physical Capacity Measurements

To determine the physical capacity of each participant, a graded exercise test (GXT) was performed using an electronically braked cycle ergometer (Kettler® DX1 Pro, Ense-Parsit, Germany). GXT is a reliable method, predominantly used to determine physical capacity. Test began at a work rate of 25 W (60 rev/min). The work rate was incremented by 25 W every 2 min until the subject could no longer maintain the required pedal cadence. Each test lasted 4–14.5 min, depending on aerobic fitness status. Peak VO_2_ was defined as the highest 15-second averaged VO_2_ obtained during the final exercise load on the test. Detailed measurements of physical capacity are described in our previous paper [14]. 

### 2.6. Blood Measurements

Blood samples for biochemical analyses were taken from a basilic vein, after overnight 12-hour fasting. In the serum samples, parameters were measured using commercially available enzyme-linked immunoassays. Total cholesterol (TC), high-density lipoprotein cholesterol (HDL-C), low-density lipoprotein cholesterol (LDL-C) and triglycerides (TG) were assessed using tests made by Siemens Healthcare Diagnostics Inc. (USA). In the plasma samples, the blood total antioxidant capacity (TAC) and thiobarbituric acid reactive substances (TBARS) were measured using tests made by CELL BIOLABS, Inc. (USA). The activity of eNOS was evaluated using a MyBioSource, Inc. (USA) kit. Concentrations of vascular endothelial growth factor (VEGF) were measured using a kit from DRG International, Inc. (USA), and C-reactive protein concentration (CRP) was measured using a marker from CELL BIOLABS, Inc. (USA). The atherogenic index of plasma (AIP) was calculated as a logarithmic ratio of plasma molar concentrations of TG and HDL-C [log TG/HDL-C], according to the method proposed by Dobiasowa and Frohlich [17].

### 2.7. Intervention

The 3-month intervention consisted of a physical exercise program involving three training sessions per week, with a total of 36 training sessions for each group. The training programs of both groups were comparable in exercise volume, and varied only in the nature of the effort (Appendix A). The detailed training program is described in our previous paper [14].

Group E underwent endurance training on cycle ergometers (Schwinn® Evolution®, Schwinn Bicycle Company, Boulder, Colorado, USA). Training sessions consisted of a 5-min low-intensity warm-up (stretching exercises; 50–60% of maximum heart rate (HR)), 45 min of training at an intensity of 60–80% of maximum HR, 5 min of non-weight-bearing cycling, finishing with 5 min of low-intensity warm-down stretching and breathing exercises (Appendix A). 

Group C underwent an endurance–strength training program consisting of a 5-min low-intensity warm-up (stretching exercises; 50–60% of maximum HR), 20 min of strength exercises using a neck barbell and gymnastics ball, 25 min of endurance exercise using cycle ergometers at an intensity of 60–80% of maximum HR, 5 min of non–weight-bearing cycling, and 5 min of low-intensity warm-down stretches and breathing exercises (Appendix A). Due to the need for regeneration of muscle power, the strength component was variable and repeated each week (see Appendix A). On Mondays, upper limb exercises were performed with a neck barbell; Wednesdays involved spine-stabilizing exercises, deep muscle-forming exercises, and balance-adjusting exercises with a gymnastic ball; on Fridays, lower limb exercises with a neck barbell were carried out. The number of repetitions in the sets was dependent on the subject’s capabilities and was equal to the number of repetitions performed correctly. Based on other studies regarding relationship between the number of repetitions and selected percentages of one-repetition maximum in free weight exercises [18] it was estimated that in our research the number of repetitions in the sets should be about 16 in barbell curls and 30 in barbell squats, that was estimated to be between 50% and 80% of their one-repetition maximum. The number of repetitions was systematically increased with the increase in subject’s muscle strength. Between sets, a 10–15 s break was introduced, during which time participants performed isometric exercises. HR during both physical trainings was monitored with a Suunto Fitness Solution® device (Suunto, Vantaa, Finland).

### 2.8. Statistical Analyses

All data are expressed as mean ± SD. All statistical analyses were performed using the STATISTICA v. 10.0 software package (StatSoft®, Krakow, Poland). Some data violated normality and demonstrated heterogeneous variability; therefore, except Student’s *t*-test, non-parametric tests were used. The Mann–Whitney U test was used to evaluate statistically significant differences between groups. The Wilcoxon rank sum test was used to assess statistically significant differences between variables before and after the 3-month intervention. Spearman’s rank analysis was used to calculate correlation coefficients. A sample size was determined according to changes in VO_2_ peak. A total of 15 subjects per group was calculated to yield at least 80% power of detecting an intervention effect as statistically significant at the 0.05 α level, with a detectable effect size of 0.8.

## 3. Results

Thirty-nine subjects underwent analysis—22 from Group E (mean age: 51 ± 8 years) and 17 from Group C (mean age: 49 ± 10 years). The compliance with prescribed intervention was 83.2%. Both groups were similar in terms of anthropometric parameters and mean age; these details are reported in our previous study [14]. 

Before intervention began, there was no correlation between the examined indicators and the age of the subjects. In order to control potential confounders, which could affect baseline data, correlation coefficients were calculated to assess relationship between body mass (kg)/fat mass content (%) [14] and other studied parameters. In the group assigned to endurance training before intervention, a correlation between body fat mass content and LDL-cholesterol concentration as well as total cholesterol was founded. In the group assigned to combined training before intervention, a correlation between body mass and TBARS concentration was founded and a correlation between body fat mass content and concentration of CRP.

Prior to the intervention, no differences between the groups—in any of the measured parameters—were observed. After the study, various changes were observed within each of the training groups but no significant differences between the groups were noted. The data for study population before and after intervention are summarized in Table 1 and Table 2.

In our previous study [14], we reported significant and comparable decreases in body mass, BMI, waist circumference (Table 1), resting HR, resting SBP and DBP following the 3-month endurance and endurance–strength training program. These changes were accompanied by an improvement in peak rate of oxygen consumption (VO_2_peak) after both interventions (Table 1). In the present study, we demonstrated that both (endurance and combined training) exert a similar, positive effect on total-body skeletal muscle mass index (Group E: *p* < 0.05; Group C: *p* < 0.01) (Table 1); however, in the case of the latter, only combined training caused significant changes in VAI (*p* < 0.05; Table 1). Both training programs led to significant decrease in TC (*P* < 0.05 for both groups) but only endurance training alone resulted in a significant increase in HDL-C concentration (*p* < 0.01) and activity of eNOS (*p* < 0.05), and a decrease in TBARS concentration (*p* < 0.05; Table 2). The combined training resulted in a significant decrease in LDL-C concentration, CRP concentration, and AIP (*p* < 0.05; Table 2). Neither endurance nor combined training resulted in significant changes in VEGF, TG or TAC concentration. Nevertheless, in both groups, TAC concentration was within the reference range of 1.10–1.54 mmol CRE·L^−1^ (Cell Biolabs, Inc.) recommended for the working European population (Table 2).

There were strong positive correlations between resting DBP and TBARS concentration in subjects before and after endurance training (*r* = 0.44; *p* < 0.05 and *r* = 0.554; *p* < 0.01, respectively; Figure 1 and Figure 2), and a positive correlation between resting SBP and TBARS concentration in subjects before the combined training was also observed (*r* = 0.783; *p* < 0.01; Figure 3).

## 4. Discussion

Different types of physical training can induce different responses, in terms of changes in metabolic characteristics [12]. Therefore, from a clinical perspective, it is important to select an intervention that is both appropriate and effective in augmenting endothelial function in individuals with obesity.

There are differences in prevalence of cardiovascular disease in men and women across the lifespan. For that reason, it is important to study endothelial function in both sexes. Our investigation focused only on women with obesity. Men did not respond to the invitation to participate in this project. Because of that, it is unknown if the training intervention would cause similar changes in the studied indicators in men of similar age. Moreover, it is likely that vasculoprotective function of the estrogen, affect observed changes in women, putting them in a better position compared with men, until menopause.

Our findings show that both endurance and endurance–strength training exert a similar, positive effect on body mass, BMI, waist circumference and VO_2_peak [14], although only combined training resulted in significant changes in VAI (*p* < 0.05). It should be noted that no significant differences between groups before and after completion of the training programs were observed. The results of the study presented herein confirm the findings of other authors [19], in that the combination of strength training and endurance training may be more effective than endurance training alone, in terms of reduction of visceral fat while maintaining/increasing muscle mass. New findings indicate that combined training may also result in a significant improvement in metabolic indicators, inter alia, lipid metabolism of people who exercise [20]. In our study, the different training sessions resulted in diverse metabolic responses; specifically, in women undergoing combined training, the post-training increase in total-body skeletal muscle index (*p* < 0.01) and reduction in VAI (*p* < 0.05) were accompanied by decreased concentrations of TC (*p* < 0.05), LDL-C (*p* < 0.05) and AIP (*p* < 0.05). Kasapis and Thompson [21] state that the effectiveness of strength exercises in training programs may result from the direct effect of this type of exercise on reduction of cytokine production in adipose tissue, muscle, and mononuclear cells and/or from the indirect effect of training on endothelial function and insulin sensitivity. The results of the present study also showed that the addition of the strength component of the training program results in decrease in CRP concentration (*p* < 0.05).

It has been reported that in the women undergoing endurance training alone, the post-training increase in total-body skeletal muscle index (*p* < 0.05) was accompanied by decreased TC (*p* < 0.05) and increased HDL-C (*p* < 0.01). Most research published to date estimates that endurance exercise can increase HDL-C by 3–9% [22]. In our study, a 9.1% increase in HDL-C was reported in women undergoing endurance training alone. It should be noted that in the group undergoing combined training, there was an 8.5% increase in HDL-C. Although not significant, it is likely that this increase influenced the reduction of AIP, defining the blood concentrations of TG and HDL-C (*p* < 0.05).

Favorable changes in the vascular system, caused by systematic exercise, are most commonly explained by inhibition of the activation of the sympathetic nervous system and the renin-angiotensin system as well as by the increase in the shear stress [23]. Under the influence of physical activity, there is an increase in blood flow in the vessel, which increases the mechanical load on the wall; in turn, this deforms the mechanosensitive channels of endothelial cells and initiates a molecular cascade leading to secretion of compounds that affect the size of the local flow. The important factors here are the functional and structural adaptive changes in the vasculature supplying blood to the skeletal muscles, especially in the microvessels [24]. Not only is shear stress relevant here, but also the transcription profile of blood flow, which maintains the phenotypic balance of the endothelial cells, providing protection against oxidative stress, inflammatory processes or apoptosis [25].

The different kinds of training programs used in the present study provided an important mechanism for normalizing blood pressure, but only endurance training resulted in increase in eNOS activity (*p* < 0.05). Most often, in the initial stages of training, functional changes in the vessels are observed, consisting of—among other things—increased NO synthesis, which results in a change in peripheral vascular resistance [26]. Studies by Gielen et al. [27] showed that during endurance training, increased blood flow in a blood vessel induces increased mRNA and protein expression, thereby leading to increased eNOS activity and NO production. Their study revealed also that endurance training improves the antioxidant capacity of the blood. The reason explained Schröder et al. [28] who demonstrated that, during training, TBARS production is increased as a result of increased production of reactive oxygen and nitrogen species (RONS) in skeletal muscle cells, erythrocytes and hepatocytes; in turn, this activates the “stocks” of antioxidants protecting the membrane lipids of blood vessels. In our study, no significant changes in TAC were observed after either training program; however, a decrease in TBARS concentration was demonstrated in the women undergoing endurance training (*p* < 0.01). 

The studies of other authors confirm that physical training can reduce the concentration of malondialdehyde (MDA), which can be quantified colorimetrically following its controlled reaction with thiobarbituric acid. This reduces the formation of superoxide anion radicals and peroxynitrite in the aorta, primarily due to increased activity of superoxide dismutase and catalase, and reduced activity of reduced nicotinamide adenine dinucleotide phosphate (NADPH) oxidase [29]. Therefore, improvement of the oxidant–antioxidant status of blood and tissues, through physical training, may be one of the mechanisms responsible for the systemic regulation of endothelial function. High levels of MDA in blood induces binding with HDL-C lipoprotein particles. In response to the formation of these modified HDL-C lipoproteins, activation of PKC-β2 protein kinase occurs, lowering eNOS phosphorylation and thereby reducing the release of NO by the endothelium [30]. Schuler et al. [31] postulate that the change in HDL-C metabolism during exercise may take place, inter alia, to bind high concentrations of MDA in the blood. In our study, in the group of women who underwent endurance training alone, a post-training increase in HDL-C and eNOS activity and decrease in the concentration of TBARS was noted, but there was no correlation between the indicators. 

Oxidative stress and endothelial dysfunction interact in a vicious circle mechanism, in which bioavailability of NO is decreased as RONS scavenge NO. A lack of NO results in a reduction in antioxidant potential, which leads to a further increase in the redox imbalance [32]. Nitric oxide is able to decrease total peripheral resistance and blood pressure [33]. Moreover, previous studies revealed that endurance aerobic exercise significantly decrease blood pressure as well as oxidative stress [34]. Our study confirm the relationship between blood pressure and oxidative stress already observed by other authors [35]. The correlation in endurance training group demonstrated a relationship between resting DBP and TBARS concentration in plasma, both before and after the training. This suggests that endurance training can reduce diastolic blood pressure due to the high bioavailability of NO in the mechanism associated with the reduction of oxidative stress. A significant reduction in blood pressure (*p* < 0.01) was observed in both endurance and combined training groups [14], however in the latter, a correlation between resting SBP and TBARS concentration was significant only before the training. It is possible that in the group of women who underwent combined training, normalization of blood pressure has been occurred independently of TBARS and eNOS concentration.

The increase in skeletal muscle blood flow induced by increased capillary network density is most often associated with episodes of tissue hypoxia that occur during physical activity [24]. During skeletal muscle contraction, shear stress and other mechanical forces occur initiating angiogenesis. Hypoxia induced by intense physical exercise is the main external factor that stimulates transcription of the gene encoding the vascular endothelial growth factor [36]. VEGF protein synthesis is the process responsible for the initial stages of creating a network of blood vessels. The manner by which the process of physical training-induced angiogenesis proceeds depends on the type of training (endurance or strength), its intensity (below or above the anaerobic threshold), and duration [37]. In our study, neither group showed changes in VEGF post-training. Levels of VEGF were compared in both groups, and did not differ significantly. The results confirm that an elevated level of VEGF contributes to angiogenesis only during the early stage of the training process. As a result of elevated levels of VEGF in the initial phase of training, recruitment of the heat-shock protein HSP90 and activation of the phosphatidylinositol 3-kinase/Akt kinase pathway may occur, which is associated with increased phosphorylation of eNOS particles and increased activity of this enzyme in the later phase of training [38]. Our observations confirm that, despite adaptive changes in the circulatory system, a post-training increase in VEGF does not occur; however, a post-training increase in eNOS activity and a decrease in plasma TBARS concentration occurs following the three-month endurance training program. 

The results of the present study suggest that the mechanisms of adaptive functional and structural changes in the vascular system may be different depending on the type of training. Although the combined training had a positive effect on lipid metabolism and significantly reduced CRP levels, only endurance training favorably changed indicators of endothelial function in women with obesity. This may be caused by retrograde blood flow associated with anaerobic exercise, occurring more frequently during endurance–strength training, which may adversely affect the endothelium [39]. It should be noted that even a short-term, high-intensity dynamic physical effort increases the retrograde shear stress, which is associated with the retrograde flow component in the blood flow pattern within the cardiac cycle, at the start of diastole [40]. In contrast, endurance training was strictly dynamic and aerobic, allowing the blood stream to increase in the vessel. Shear stress, occurring in such circumstance, by increasing the release of nitric oxide, augments endotelium-dependent vasodilation and inhibits multiple processes involved in atherogenesis [41]. Authors suppose that different impact of trainings on eNOS activation might be associated with a longer stimulus effect in the form of endurance exercises during a three-month intervention compared to a group undergoing combined training (45 versus 25 minutes). It is likely that a 25-minute endurance component in a combined training is an insufficient stimulus to initiate changes in the vascular endothelium. However, it is also possible that vasodilatation functions of the endothelium had been improved regardless of changes in eNOS activity in women undergoing combined training, as the intervention may have contributed to an increase of sensitivity of vascular smooth muscles to other endothelial vasodilatation factors, not measured in this paper.

It is probable that it is not only the type of the training, but also the volume and choice of exercise that plays a role in inducing patterns of shear forces in the blood vessels, which impact the vascular endothelial function. Williams et al. [42] demonstrated that changes in the cardiovascular system resulting from strength training are similar to the effects of endurance training when using a small load and a higher number of repetitions. That is why, analyses evaluating the effect of endothelial function under the influence of different patterns of shear stress, as in our own studies, may be inconclusive and still need to be clarified. We acknowledge that small sample size (44 enrolled, 39 completed) limits our ability to draw solid conclusions about the relative efficacy of the endurance–strength training program versus the endurance training program in the women with obesity

## 5. Conclusions

Overall, our results show that both the three-month endurance and endurance–strength training programs led to the improvement of lipid metabolism, but only endurance training favorably changed indicators of endothelial functions in women with obesity. The present study highlights the need for more research comparing the efficacy of combined and endurance trainings. It is still not known exactly what proportions of endurance to strength training in combined modalities are most effective to restore endothelial function. Future studies should take into consideration larger study group, participation of men in the research and evaluation of both biochemical and functional markers associated with endothelial function.

## Figures and Tables

**Figure 1 ijerph-16-04291-f001:**
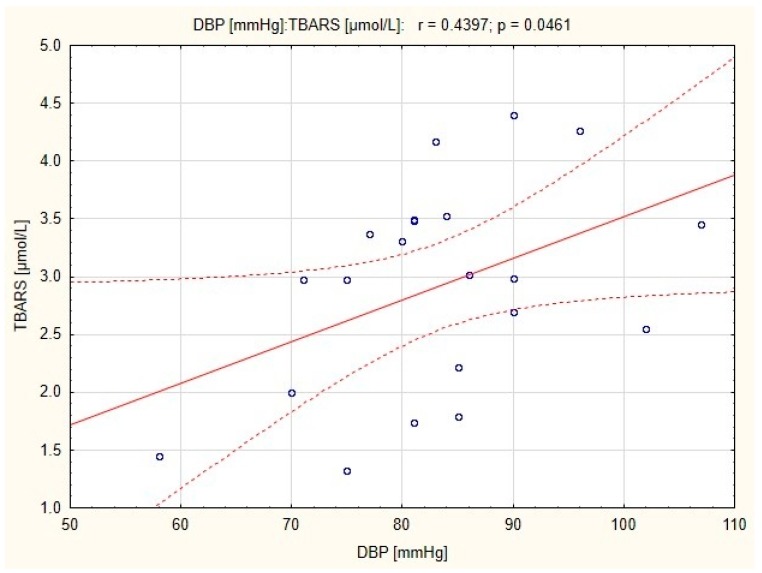
Correlation between resting diastolic blood pressure (DBP) and thiobarbituric acid reactive substances (TBARS) concentration in women with obesity before endurance training.

**Figure 2 ijerph-16-04291-f002:**
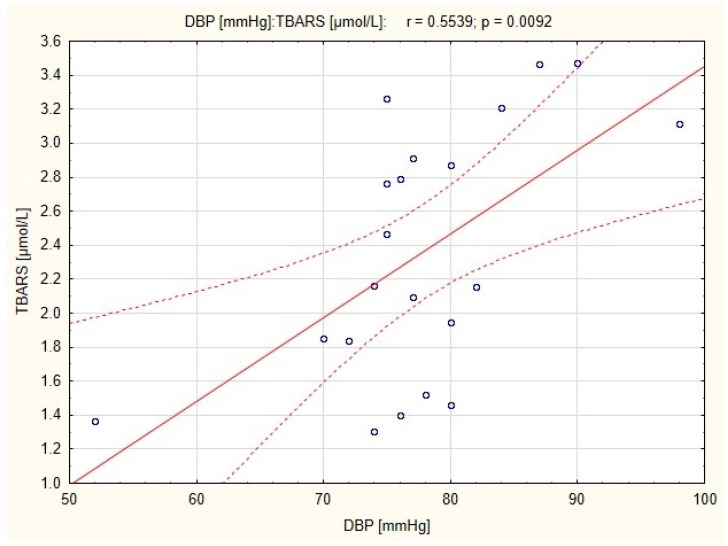
Correlation between resting diastolic blood pressure (DBP) and thiobarbituric acid reactive substances (TBARS) concentration in women with obesity after endurance training.

**Figure 3 ijerph-16-04291-f003:**
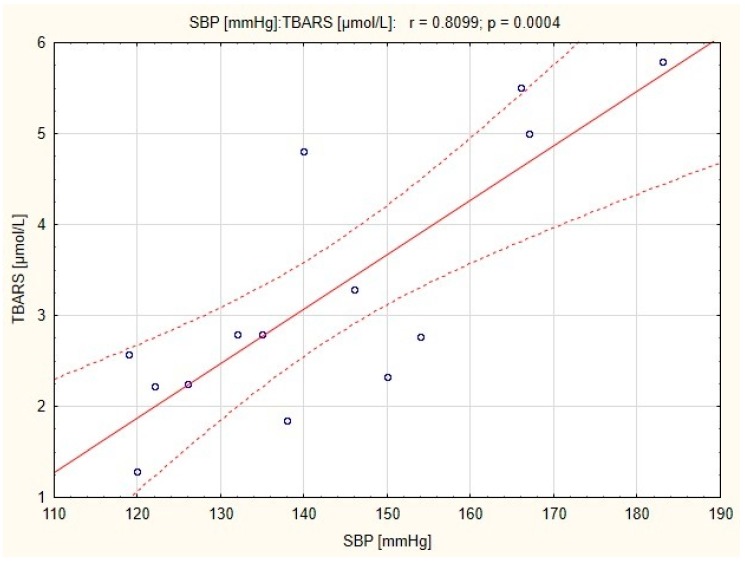
Correlation between resting systolic blood pressure (SBP) and thiobarbituric acid reactive substances (TBARS) concentration in women with obesity before the combined training.

**Table 1 ijerph-16-04291-t001:** Anthropometric parameters of E and C groups before and after a 3-month training programs.

	ENDURANCE TRAINING (E)			COMBINED TRAINING (C)		
	Before Intervention	After Intervention		Before Intervention	After Intervention	
	mean ± SD	mean ± SD	*P*	mean ± SD	mean ± SD	*P*
Body mass (kg) *	94.6 ± 17.9	92.2 ± 17.4	<0.01	93.9 ± 13.3	91.3 ± 13.5	0.01
Body mass index (kg/m^2^) *	35.9 ± 5.2	35.1 ± 5.1	<0.01	35.0 ± 3.9	34.1 ± 4.2	<0.01
Waist circumference (cm) *	111.5 ± 10.6	106.2 ± 11.4	<0.01	111.2 ± 11	103.1 ± 9.6	<0.01
Visceral adiposity index	2.6 ± 1.4	2.3 ± 1.5	NS	2.2 ± 1.3	1.9 ± 1.5	<0.05
Total-body skeletal muscle mass index (kg)	21.9 ± 2.7	22.4 ± 3.2	<0.05	22.1 ± 2.5	23.3 ± 3.2	<0.01
VO_2_peak [ml/(kg×min)] *	17.3 ± 2.1	20.4 ± 3.3	<0.01	18.3 ± 3.4	22.3 ± 4.7	<0.01

Data are presented as mean ± SD. No significant differences between groups were observed at any time-point. *P* values indicate statistically significant differences in studied parameters between baseline and month 3. * Data according to Reference [14].

**Table 2 ijerph-16-04291-t002:** Biochemical parameters of E and C groups before and after a 3-month training programs.

	ENDURANCE TRAINING (E)			COMBINED TRAINING (C)		
	Before Intervention	After Intervention		Before Intervention	After Intervention	
	mean ± SD	mean ± SD	*P*	mean ± SD	mean ± SD	*P*
eNOS (ng/mL)	26.1 ± 25.2	29.9 ± 22.5	<0.05	32.7 ± 26.5	34.4 ± 27.0	NS
VEGF (pg/mL)	107.1 ± 47.4	110.8 ± 82.9	NS	113.7 ± 55.0	128.7 ± 80.4	NS
TBARS (µmol/L)	2.91 ± 0.90	2.35 ± 0.7	<0.01	3.23 ± 1.40	2.63 ± 1.00	NS
TAC (mmolCRE/L)	1.31 ± 0.20	1.29 ± 0.20	NS	1.24 ± 0.30	1.26 ± 0.20	NS
Total cholesterol (mmol/L)	5.65 ± 1.03	5.30 ± 0.98	<0.05	5.85 ± 1.07	5.52 ± 1.09	<0.05
HDL-cholesterol (mmol/L)	1.33 ± 0.36	1.45 ± 0.34	<0.01	1.44 ± 0.56	1.56 ± 0.49	NS
LDL-cholesterol (mmol/L)	3.39 ± 0.80	3.16 ± 0.88	NS	3.59 ± 0.70	3.30 ± 0.77	<0.05
Triglycerides (mmol/L)	1.51 ± 0.62	1.54 ± 0.75	NS	1.33 ± 0.52	1.31 ± 0.58	NS
AIP	0.03 ± 0.26	-0.01 ± 0.24	NS	− 0.04 ± 0.27	− 0.09 ± 0.27	<0.05
CRP (mg/L)	4.18 ± 2.50	3.45 ± 2.50	NS	3.49 ± 3.20	2.52 ± 1.90	<0.05

Data are presented as mean ± SD. No significant differences between groups were observed before and after the study. *P* values indicate statistically significant differences in oxide studied parameters between baseline and month 3. eNOS, endothelial nitric synthase; VEGF, vascular endothelial growth factor; TBARS, thiobarbituric acid reactive substances; TAC, total antioxidant capacity; AIP, atherogenic index of plasma; CRP, C-reactive protein.

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
