# Peer review of "Effects of Endurance and Endurance–Strength Training on Endothelial Function in Women with Obesity: A Randomized Trial"

_ijerph, 2019, doi:10.3390/ijerph16214291_

Round 1
Reviewer 1 Report
The manuscript entitled “Effects of endurance and endurance–strength training on endothelial function in women with obesity: A randomized trial” represents a prospective randomized trial assessing the effect of combined endurance-strength training vs. endurance training alone for three months on endothelial function, lipid metabolism, and risk of atherosclerosis in obese women. The study extended the findings from the authors’ previous work by demonstrating a positive effect on total-body skeletal muscle mass index in both groups, changes in HDL-C, eNOS, and TBARS in the endurance group, changes in VAI, LDL-C, AIP, and CRP in the combined group.
My comments are as follows:
Methods, Page 4: “A sample size of at least 15 subjects in each group was calculated to yield at least 80% power of detecting an intervention effect as statistically significant at the 0.05 α level”. Please clarify the magnitude of between-group difference that was defined as an intervention effect.
The rationale for examining the correlation between resting systolic or diastolic blood pressure and TBARS should be clarified. It is unclear why only these two specific parameters were analyzed. The discussion on this finding should be expanded.
Significant changes in HDL-C, eNOS, and TBARS were only observed in the endurance training group but not in the combined endurance-strength training group. These changes cannot be attributed to the endurance training, as the combined group also included endurance training sessions. To avoid confusion, it is advisable to rephrase the conclusion that “only endurance training alone favorably changed indicators of endothelial functions in women with obesity”.
Reviewer 2 Report
1 – The novelty is not clear. Why endurance training alone or combined training would elicit different physiological responses than what it is already shown on literature?
2 – How the 1-RM was assessed?
3 – It is possible that the training became less efficient due to no re-evaluation? Specially in the endurance group.
4 – The sample size was calculated using which effect size?
5 – It is not clear which statistical test was applied for the data in Tables 1 and 2. The authors have a 2x2 design, multiple t-tests (or Mann-Whitney) are not the adequate test for this situation.
6 – Limitations should be in discussion, and definitely not right before conclusion.
7 – It is not clear the relevance of the TBARSxDBP correlation, neither in baseline or post-training.
8 – Authors extensively discuss that the endurance group improved endothelial function through eNOS activation and higher NO bioavailability. However, the combined training group also performed endurance training (with lower volume, of course). What is the possible mechanisms the could have affected the no-improvement of endothelial function in this group?
Reviewer 3 Report
Affiliations
Please check the size of the words of the affiliation 1 to be consistent with the remaining affiliations
ABSTRACT
Line 20, there is a typo error. Sentence starts with “n” PLEASE remove or reword it.
Please include in the results section at least the p value of the significant differences or non-significant differences found.
Keywords: authors should include other different words than in the title (i.e. strength training) in order to wider the search for researchers.
Introduction
The introduction is well-structured, short but concise. However, I am missing a more specific, if its possible, of a detailed physical exercise programme. Authors stated short-term training, but how many days per week? What intensity? What type of training?
Methods
Study participants – Author should include information about causes of dropouts. If they finally enrolled 44 women, what happened with the remaining women interested?
Please state more information, although briefly, about the test applied. Although authors cite a previous paper, a brief and general information should be included
Specify also the number of hours required for fasting
Can you include as supplementary analyses an example of each of the trainings exposed (e and c)?
What software the authors used for calculation power?
Results
How the age is influencing the results?
Why the authors did not control for potential confounders? Data can be affected by baseline starting data. If the authors have done previously a confounder analyses, they should also state it as a good point for quality.
Have the authors made changes for those values that could be considered as outliers? (ie. Winsorization method,?)
Discussion
Discussion is well-structured and very complete explaining the mechanism that could happen after the changes obtained.
Authors should include either in the introduction or discussion the importance or relevance of only selecting women with obesity. Does the exercise changes could be due to sex of participants?
Lastly, please include in the discussion the limitation sentence and start the conclusion with overall findings.
Include after conclusion a sentence with further studies ideas
Round 2
Reviewer 2 Report
No further comments.
Reviewer 3 Report
I have no further comments to the authors.